EcoCountHelper: an R package and analytical pipeline for the analysis of ecological count data using GLMMs, and a case study of bats in Grand Teton National Park

http://orcid.org/0000-0003-1465-1819 Cole Hunter J. hunterjcole25@gmail.com
http://orcid.org/0000-0002-2642-3728 Gomes Dylan GE
http://orcid.org/0000-0003-3084-2973 Barber Jesse R.
Department of Biological Sciences, Boise State University , Boise, ID , United States
Silva José Maria
Electronic publication date: 2022 Dec 14
Publication date: 2022
Volume: 10
Electronic Location ID: e14509
Received 2022 Mar 1; Accepted 2022 Nov 13
Copyright: © 2022 Cole et al.
Copyright year: 2022
Copyright holder: Cole et al.
License: This is an open access article distributed under the terms of the Creative Commons Attribution License, which permits unrestricted use, distribution, reproduction and adaptation in any medium and for any purpose provided that it is properly attributed. For attribution, the original author(s), title, publication source (PeerJ) and either DOI or URL of the article must be cited.
License URL: https://creativecommons.org/licenses/by/4.0/

Keywords: Generalized linear mixed effects model, Visualization, Abundance, CRAN, R package

Funding: Grand Teton National Park UW-NPS Research Station The work was conducted under the auspice of Grand Teton National Park and funded under a cooperative ecosystem study unit agreement and also received funding from the UW-NPS Research Station. The funders had no role in study design, data collection and analysis, decision to publish, or preparation of the manuscript.

==============================
Here we detail the use of an R package, ‘EcoCountHelper’, and an associated analytical pipeline aimed at making generalized linear mixed-effects model (GLMM)-based analysis of ecological count data more accessible. We recommend a GLMM-based analysis workflow that allows the user to (1) employ selection of distributional forms (Poisson vs negative binomial) and zero-inflation (ZIP and ZINB, respectively) using AIC and variance-mean plots, (2) examine models for goodness-of-fit using simulated residual diagnostics, (3) interpret model results via easy to understand outputs of changes in predicted responses, and (4) compare the magnitude of predictor variable effects via effects plots. Our package uses a series of easy-to-use functions that can accept both wide- and long-form multi-taxa count data without the need for programming experience. To demonstrate the utility of this approach, we use our package to model acoustic bat activity data relative to multiple landscape characteristics in a protected area (Grand Teton National Park), which is threatened by encroaching disease—white nose syndrome. Global threats to bat conservation such as disease and deforestation have prompted extensive research to better understand bat ecology. Notwithstanding these efforts, managers operating on lands crucial to the persistence of bat populations are often equipped with too little information regarding local bat activity to make informed land-management decisions. In our case study in the Tetons, we found that an increased prevalence of porous buildings increases activity levels of Eptesicus fuscus and Myotis volans; Myotis lucifugus activity decreases as distance to water increases; and Myotis volans activity increases with the amount of forested area. By using GLMMs in tandem with ‘EcoCountHelper’, managers without advanced programmatic or statistical expertise can assess the effects of landscape characteristics on wildlife in a statistically-robust framework.

Introduction

Wildlife managers are often tasked with understanding ecological relationships that may shed light on the potential effects of proposed management actions. While these relationships may vary in complexity and scale, a common means of assessing interactions between wildlife and characteristics of the environments they occupy is to count some metric of wildlife abundance or activity, then quantitatively relate those counts to environmental characteristics of interest. Examples of this ecological relationship assessment technique include examination of bat activity at varying distances from a road (Berthinussen & Altringham, 2012), assessment of bird abundance with respect to river noise levels and other habitat characteristics (Gomes et al., 2021), and examining the effect of seasonality and weather on sika deer activity (Ikeda et al., 2015). Perhaps one of the most widely available, flexible and powerful analyses managers can use to gauge the potential effects of management decisions on wildlife are generalized linear mixed models (GLMMs).

While the process required for a statistically-sound GLMM-based analysis is well documented (Bolker, 2008; Kéry & Royle, 2015; Harrison et al., 2018; McElreath, 2020), implementing that process for a specific dataset can be tedious and difficult without guidance. Additionally, the lack of standardization in how analyses are performed can lead to variability in statistical inference (Silberzahn et al., 2018). The potential for GLMMs to be inaccessible to managers is unfortunate, as GLMM analyses can be relatively simple in execution, flexible, and allow structured (hierarchical) data to be accounted for in statistics. GLMMs are also robust to imbalanced data, are more flexible in their assumptions than many other analytical methods and can accommodate both categorical and numeric data. These models can be particularly useful for analyses involving count data with repeated measures at multiple locations (e.g., acoustic monitoring bat activity, aggregated telemetry detections, avian point counts, camera trapping) (Bolker et al., 2009). To reduce the coding skills necessary for GLMM-based analyses and guide users through the GLMM workflow, we developed the R package ‘EcoCountHelper’. Here, we have two aims: (1) detail the function and use of the R package, ‘EcoCountHelper’, that we developed to aid in the analysis of our multispecies count data, and (2) describe a simple GLMM analysis of multi-species bat activity data using ‘EcoCountHelper’.

Our case study involves archetypal count data, bat acoustic monitoring data, collected in Grand Teton National Park. A crown jewel of the US protected area network, Grand Teton National Park lies at the southern end of the Greater Yellowstone Ecosystem—an area dominated by high elevation coniferous forests and sage steppe plains. In March of 2016, the Washington Department of Fish and Wildlife found bats afflicted by white-nose syndrome in King County, WA (Haman, Hibbard & Lubeck, 2016). The arrival of the deadly Pseudogymnoascus destructans fungus in a Western state, 10 years after it was first introduced to North America in New York (White-nose Syndrome Response Team, 2020), increased the imminence of the threat posed by white-nose syndrome to western bat populations and set a precedent to better understand habitat characteristics that may be managed to help bolster bat populations in the face of a looming epidemic.

Prior to our project, multiple bat surveys were conducted in Grand Teton (Genter & Metzgar, 1985; Keinath, 2005), however they were primarily descriptive in nature and did not aim to quantify the importance of landscape features to the bats they were surveying. In our attempts to quantify the importance of different habitat characteristics to bats in Grand Teton, we predicted that bat space use would be driven by several factors including elevation (Cryan, Bogan & Altenbach, 2000), distance to water (Evelyn, Stiles & Young, 2004), land cover type (Evelyn, Stiles & Young, 2004; Russo & Ancillotto, 2015), proportion of porous buildings that could serve as day roosts (Voigt et al., 2016), lunar phase (Saldaña-Vázquez & Munguía-Rosas, 2013), ordinal date (Weller & Baldwin, 2012), and the presence of non-natural light sources (Stone, Harris & Jones, 2015). To advance future work on bat population biology and make modelling bat activity more accessible, we describe an analytical pipeline using ‘EcoCountHelper’ to facilitate the identification of the most appropriate error distribution, assessment of zero-inflation, goodness-of-fit testing, and model interpretation processes associated with analyses incorporating generalized linear mixed models (GLMMs).

Materials and Methods

GLMM workflow

We have adapted a workflow for constructing and interpreting GLMMs based on the ‘glmmTMB’ package (Bolker et al., 2012; Bolker, 2016; Brooks et al., 2017b). An outline of this adapted workflow involves the following steps for each taxonomic group of interest: We first decide on predictor variables and random effects structures for a response of interest.

We then compare the fits of Poisson and negative binomial error distributions (as well as the potential for zero-inflation) by corroborating AIC values, mean-variance plots, and examining simulated residual test plots—the latter of which also help identify outliers and uniformity.

We then interpret model results through both standardized coefficient visualizations and predicted changes to response variables, given a user-specified increase in a single predictor variable.

To aid researchers, and particularly land and wildlife managers, in implementing GLMM-based analyses for count data, we developed the ‘EcoCountHelper’ R package to simplify this workflow.

Package workflow

The ‘EcoCountHelper’ package (downloadable via Cole, Gomes & Barber, 2021a, 2021b) is meant to assist researchers for the portions of a GLMM-based analysis after the point at which candidate models have been generated. It is important that individuals using ‘EcoCountHelper’ take care to carefully consider candidate model structures, and to appropriately implement GLMM components such as zero-inflation formula (Martin et al., 2005) and random effects (Harrison et al., 2018; Gomes, 2022). Despite the popularity of selecting predictor variables for both conditional model zero-inflation formulas via dredging with information criteria (e.g., AIC), there are many pitfalls with using these methods (Guthery et al., 2005; Link & Barker, 2006; Bolker, 2008). While there are multiple excellent resources for learning about GLMMs (Bolker, 2008), Harrison et al. (2018) is an excellent touchstone for these analyses. Once one has already generated appropriate candidate models with ‘glmmTMB’ (Bolker et al., 2012; Bolker, 2016; Brooks et al., 2017a) and prepared all data (see Supplemental “EcoCountHelperExample” vignette), the workflow we suggest for GLMM analyses using the ‘EcoCountHelper’ functions (Table 1) can be executed.

Table 1 EcoCountHelper function purposes.

Analytical framework component	Associated function(s)	
Choose an error distribution family
Are data zero-inflated?	ModelCompare, DistFitLong, & DistFitWide	
Goodness-of-fit diagnostic plots/tests	ResidPlotLong & ResidPlotWide	
Examining relative effect sizes	EffectsPlotter	
Interpreting scaled estimates	RealEffectText, RealEffectTabLong, & RealEffectTabWide	
Note:

Each step in the analytical framework we outline in this document can be facilitated by one or more EcoCountHelper functions.

The first step in our pipeline is to select the best-fitting model for each response group under investigation, taxonomic (e.g., species, genus) or otherwise (e.g., indistinguishable grouped taxa, foraging guilds, operational taxonomic units). Two non-exclusive techniques for deciding whether to use zero-inflated models and selecting the best error distribution family for a particular count dataset are: (1) examine AIC values to assess zero-inflation and select an error distribution, and (2) examine the mean-variance relationship within the data and determine an error distribution family that best mirrors the mean-variance relationship. The ‘EcoCountHelper’ package has a function, “ModelCompare”, for simultaneously obtaining AIC values for each group’s candidate models and creating a vector of the models with the lowest AIC scores for each group. The “DistFit” family of functions aids in visual examinations of mean-variance relationships for each group in the analysis. These “DistFit” functions allow users to specify vectors by which data should be aggregated for examining the mean-variance relationship of the data. The “DistFit” functions then generate a scatterplot displaying the mean-variance relationship for the data and draw lines through the scatterplot showing three common error distributions used for count data: Poisson, negative binomial with a linear mean-variance relationship (a.k.a. quasi-Poisson), and negative binomial with a quadratic mean-variance relationship (variance increases by the squared mean) (Bolker et al., 2012). The functions used during the model construction process that correspond with each of these error distributions are “poisson” from the ‘stats’ package, “nbinom1” from ‘glmmTMB’, and “nbinom2” from ‘glmmTMB’, respectively. For the sake of simplicity, we will use these function names to describe the associated distributions from this point on. After generating mean-variance plots using the “DistFit” family of functions, the user can then choose the model with the most appropriate error distribution for each group by visually examining the plots and choosing the model employing the error distribution that best fits. By using both of these methods (AIC and mean-variance relationships) we generally arrive at the same conclusion about which distribution best fits the data. There is, however, some subjectivity in distribution choice when both Poisson and negative binomial models appear to fit similarly. In our experience with this situation, the differences in parameter estimates are usually negligible, and thus distributional choice will likely be inconsequential. In this case, one may prefer the simpler Poisson distribution because there are fewer parameters to estimate (Poisson is a special case of negative binomial model in which the variance equals the mean, so there is only one term to estimate instead of two). It is also worth noting that using AIC values to determine whether zero-inflation formulas result in better model fit does not test for zero-inflation of data. Rather, the use of AIC values to compare candidate models either employing or omitting zero-inflation formulas provides information about whether variables specified in zero-inflation formulas explain any zero-inflation that does exist within given data. Zero-inflation formulas may be implemented to examine the effects of one or multiple predictors on zero-inflation of data, but care must be taken to ensure that predictors specified in zero-inflated formulas could reasonably contribute to zero-inflation of data.

Following the model selection process, it’s important to test the goodness-of-fit of the chosen models. Because a model fits a dataset better than all others does not mean it is adequate for making inferences or predictions. Plotting residuals (or simulated residuals) is a quick and often useful step in model criticism, but it is important to note that there are many other ways to assess goodness-of-fit. EcoCountHelper’s “ResidPlot” family of functions provide a simple way to simulate residuals for models that employ a non-Gaussian error distribution and generate plots to visually check residual uniformity, outliers, and over-/under-dispersion (important indicators of goodness of fit). These functions borrow functions from the ‘DHARMa’ R package (Hartig, 2022). It is important to note that if precise predictions are required from a model in order to make management decisions, the “gold-standard” for assessing the predictive capacity of a model is leave-one-out (a variant of k-fold) cross validation (Hawkins, Basak & Mills, 2003; Vehtari, Gelman & Gabry, 2017). Holding out part of a dataset to assess predictive capacity may be a computationally less intense alternative (Kim, 2009). These methods involve additional computational steps and are outside the scope of this article.

Assuming selected models, for all species or groups of interest, fit the data adequately, the next step is interpretation of model results by visualizing model coefficients and confidence intervals. The “EffectsPlotter” function generates coefficient plots with up to three user-specified confidence intervals surrounding each coefficient value. While many managers and practitioners are most often familiar with p-values (and an alpha of 0.05), any confidence interval chosen is arbitrary and we suggest thinking deeply about the consequences of type I and type II errors in your system and adjust confidence intervals accordingly. See the “Model Interpretation” section of the methods for additional discussion surrounding confidence interval selection. More importantly the effect size, on the scale of the original response variable, should be taken into consideration when making management decisions (Sullivan & Feinn, 2012). Because the coefficients of fitted GLMMs (except for those using an identity link, as typically used in linear (Gaussian) models) are a product of a link transformation (via the link specified during the model fitting process; with Poisson and negative binomial this is typically a log link), the coefficients cannot be used to predict the effect without back-transformation. Additionally, continuous data used during model fitting is often standardized for computational gains, increased model convergence, and comparison purposes (Gelman, 2008) which further complicates the process of making predictions with meaningful units. To simplify this process, we created the “RealEffects” family of functions in ‘EcoCountHelper’ which allows users to rapidly assess the response to given changes in predictors using untransformed “real world” values and outputs to a readable text. A flowchart integrating both conceptual and programmatic steps in the workflow proposed here can be found in Fig. 1.

Figure 1 Conceptual and applied workflow of EcoCountHelper.

An illustration of the workflow associated with GLMM analyses using EcoCountHelper. Blue boxes represent conceptual steps, and orange boxes represent the application of functions to accomplish related conceptual steps.

‘EcoCountHelper’ also has multiple accessory functions that do not directly pertain to the model fitting, model selection, and result interpretation process outlined above. One noteworthy accessory function is “scale2”. This function is identical to the base R scale function in that it standardizes a vector, but rather than subtracting the mean from each value and dividing by the standard deviation, “scale2” subtracts the mean and then divides each value by two standard deviations (Eq. (1)) as suggested by Gelman (2008). During the model fitting process, using the “scale2” function puts continuous values on the same scale as binary categorical variables allowing a direct comparison of standardized continuous coefficients and categorical coefficients.

(1) x−x¯_2σ

Package limitations

The ‘EcoCountHelper’ package was designed to make GLMMs more accessible to researchers and managers who otherwise may not have sufficient programmatic skills to carry out analyses in a timely, reproducible, and statistically responsible manner. Because many of our package’s functions are designed to return specific results (e.g., plots, test statistics) while minimizing unguided data preparation, users must adhere to relatively strict object naming schemes to ensure that pattern recognizers within the functions can identify relevant objects in the global environment. This forces users to conform to a model naming scheme that provides information about the taxonomic group the model belongs to, and also creates intermediate objects that have little or no flexibility in naming conventions. While it makes some aspects of the package inflexible, this strict naming convention allows users unfamiliar with regular expressions to forgo the exercise of identifying objects in the global environment and adding them to a vector for further use either manually or through regular expressions. Because functions that generate plots require virtually no user input for plot construction, it can be difficult to make edits to resulting plots. Again, while this may be inconvenient for programmatically savvy individuals, this inflexibility in data visualization allows for programmatically inexperienced individuals to carry out GLMM based analyses in a simple and straightforward manner.

Case study methods

Throughout the summers (June to September) of 2016 and 2017, we monitored bat activity throughout Grand Teton National Park, Wyoming, USA (referred to as Grand Teton from here on) using passive acoustic monitors. Grand Teton is composed of a relatively flat, high elevation (average ~2,073 m (National Park Service, 2019)) valley bounded by the Teton mountain range to the west, and the Gros Ventre mountain range to the east. The majority of our work occurred within the valley of the park.

To assess the impact of buildings and artificial light on the bats of Grand Teton, we used Wildlife Acoustics SM2BAT and SM3BAT units to record bat echolocation at 36 sites throughout the park with varying levels of anthropogenic infrastructure. To obtain high quality recordings, we set sampling rates for all ultrasonic recorders to 384 kHz allowing us to capture frequencies up to 192 kHz. We programmed all recorders to allow ultrasonic recordings from thirty minutes before sunset to sunrise, and to begin recording when triggered by a 16 kHz or greater signal that was 18 dB or more above background sound levels. This triggered recording scheme generates WAV files with individual bat call sequences, allowing for rapid and automated processing using call identification software (e.g., SonoBat, Kaleidoscope).

We chose monitoring sites to capture variation in multiple landscape characteristics including elevation, distance to water, land cover, and human infrastructure that may influence habitat suitability for bats (e.g., porous buildings that could serve as day roosts (Voigt et al., 2016)), the presence of non-natural light sources (Stone, Harris & Jones, 2015). In total, we monitored bat activity at 36 sites throughout the park (see Fig. 2), deploying units at 26 sites in 2016, and 27 sites in 2017 (17 of which were sites established in 2016). We deployed each acoustic monitor for five to six nights at each site during 2016 and 13 to 14 nights in 2017 due to an increased availability of units. Our sampling effort totaled 840 site-nights for both years of data collection, with 276 site-nights in 2016 and 564 site-nights in 2017. For additional information regarding data collection and preparation, see the Supplemental Information.

Figure 2 Map of the study area.

Our research was conducted throughout the valley of Grand Teton National Park. The data included in our analyses are from the 36 sites shown in this map, with the color of each site indicating the year(s) that data were collected. Map data©2015 Google.

Analysis

We used generalized linear mixed models (GLMMs) to assess the effect of the predictors mentioned above on activity for each species of bat that was detected during at least 50 site-nights (Eptesicus fuscus, Lasiurus cinereus, Lasionycteris noctivagans, Myotis evotis, Myotis lucifugus, Myotis volans and Myotis yumanensis). We ran all models in R using the ‘glmmTMB’ package (Brooks et al., 2017a) and standardized all continuous variables using our “scale2” function using the formula.

For each bat species under investigation, we generated six models (three different distribution families and a zero-inflated version of each; see below). All models shared a common conditional model structure that included year, lunar phase, ordinal date, elevation, roost-building density index, proportion of cool lights within 50 m, summed brightness score of lights within 50 m, distance to the nearest water source, proportion of developed land cover within 50 m, and proportion of forested area within 50 m as fixed effects, and monitoring site as a random intercept term. Additionally, all models used a log link function. For each species, two models were generated for each error distribution implemented in the mean-variance plots produced by the “DistFit” family of functions (“poisson”, “nbinom1”, and “nbinom2”): one with a zero-inflation formula, and one without. We modelled zero-inflation with ordinal date and site as predictors.

For each species, we determined the most appropriate error distribution by corroborating results from the synergistic “DistFit” family of functions and the “ModelCompare” function (see Package Workflow above). We determined a priori that in the case of conflicting results from the two processes above, we would rely on visual mean-variance plots for assessing a best-fit model for the proceeding analytical steps. We also used the AIC values generated by the “ModelCompare” function to determine whether a model with a zero-inflated component was necessary. Note that only the parameters specified in the zero-inflation formula are assessed for effects on zero-inflation, and while zero-inflation may exist in one’s data, ZIP or ZINB candidate models may not result in better model fit if predictors influencing zero-inflation are not specified in a zero-inflation formula. The best-fitting model for each species was subsequently tested for goodness-of-fit using EcoCountHelper’s “ResidPlot” functions. We also checked variance inflation factors (VIF) in R using the function “check_collinearity” in the ‘performance’ package (Lüdecke et al., 2020).

Model interpretation

We first examined the scaled coefficient estimates by plotting each estimate with confidence intervals on a common scale using EcoCountHelper’s “EffectPlotter” function. These plots were used to inform our understanding of the relative importance of landscape features to species-specific bat activity. While informative in terms of relative effect sizes and confidence levels surrounding those effects, these plots do not provide meaningful absolute effect sizes that can be interpreted in an ecological context. The purpose of examining effects plots of scaled and transformed coefficient estimates is to develop an understanding of the relative magnitude of effect each predictor of interest has on a bat species’ activity throughout the study area.

Rather than creating and examining effects plots, one might want to simply examine the p-values of coefficient estimates to determine which predictors warrant further investigation. We suggest avoiding making any decisions regarding the importance of landscape features for bat conservation using p-values alone. While there is nothing inherently wrong with this approach, relying on p-values alone is insufficient for meaningful interpretations of results (Sullivan & Feinn, 2012; Lin, Lucas & Shmueli, 2013; Halsey et al., 2015; Vidgen & Yasseri, 2016). By examining scaled coefficient estimates, one can quickly assess both the relative magnitude of, and confidence in, effect sizes. It is important for ‘EcoCountHelper’ users to determine confidence intervals and effect sizes that are suitable for their purposes. Many researchers have traditionally used 95% confidence intervals to determine whether the effect of a predictor is meaningful, but any confidence interval can be used to arbitrate the validity of an effect so long as the researcher has considered the consequences of type I errors due to a low confidence threshold. In interpreting the results of our case analysis, we examined both 85% and 95% confidence intervals. While not as conservative as 95% confidence intervals, 85% confidence intervals still provide information regarding trends in data that, given more sampling or follow-up research, could be confirmed at a higher confidence level (Cumming & Finch, 2005). Similarly, one may observe a large effect size but have insufficient data to obtain 95% confidence intervals that do not overlap zero. By tempering the interpretation of results by acknowledging the confidence surrounding those estimates, managers can act on the information available if necessary while also recognizing the need for additional research on the topic of interest. Considering the low risk posed by a type I error (i.e., persistence of historic and porous structures throughout the park) and the potential benefit of increasing our understanding of bat ecology in Grand Teton with relatively high certainty, we decided interpreting trends present at approximately the 85% confidence level was acceptable. Ultimately we suggest that researchers and managers implementing our suggested workflow do not treat it as a means of hypothesis testing, but rather as a path toward understanding a system surrounded by a level of uncertainty. Both the estimates provided by models as well as the associated uncertainty should be examined within the context of the research objectives to inform one’s conclusion about a predictor’s influence on an ecological process.

After identifying landscape features that influence activity levels, we examined the real-world effects of our predictors by calculating the factor by which bat activity changes relative to a percentage change in each predictor and associated confidence intervals. In the case of models implementing a (natural) log-link, this entails back-transforming the coefficient estimate by first unscaling and then exponentiating Euler’s number (e) by that quotient/value. Then the resulting term is exponentiated by the difference of interest in the values of the predictor. For example, if one is interested in the relative change across 50 m of elevation change, this difference ( δx) would be 50 (since the rest of the term has already been back-transformed to the original units and represents 1 unit change; in this example 1 m). Because we scaled continuous variables using two standard deviations, we used Eq. (2) to calculate the factor by which the response variable changes given a change in a predictor.

(2) (eβ2σ)δx

In Eq. (2), β represents the model coefficient of interest, σ represents the standard deviation of the unstandardized predictor, e represents Euler’s number, and δx is the difference of interest in predictor x. To facilitate these calculations, the ‘EcoCountHelper’ package contains a “RealEffect” group of functions that back-transform estimates and return unscaled response for specified changes in predictors.

Results

For the seven species of bats for which we constructed models (see Table 2), all exhibited model convergence for at least one candidate model. Corroboration of mean-variance plots and AIC values for all candidate models indicated that data were not zero-inflated and the “nbinom2” error distribution family best fit the data for all species modelled. There were no conflicting results regarding mean-variance plots and AIC values. All models exhibited adequate goodness-of-fit as determined by examining diagnostic plots, and no model parameters exhibited excessive VIF values (James et al., 2013, pp. 101–102), though the parameter capturing a site’s summed light brightness scores showed moderate collinearity (VIFs between 5.04 and 5.76) in models for E. fuscus, M. volans, and M. yumanensis.

Table 2 Species-level model results.

Species	Value	Brightness index	Prop. cool lights	Prop. developed	Structure index	Elevation	Moon illum.	Ordinal day	Prop. forest	Water distance	Year (16–17)	
Epfu	Estimate	−0.299	0.848	−1.055	−0.767	1.795	0.194	−0.159	0.765	0.401	−0.571	
SE	0.257	1.203	0.744	1.015	0.529	0.545	0.132	0.17	0.504	0.504	
p	2.43E−01	4.81E−01	1.56E−01	4.50E−01	6.98E−04	7.21E−01	2.27E−01	6.45E−06	4.26E−01	2.57E−01	
Laci	Estimate	0.696	0.536	−0.698	−0.284	0.654	0.527	0.056	2.65	−0.438	−0.427	
SE	0.217	0.936	0.612	0.772	0.448	0.445	0.109	0.161	0.414	0.404	
p	1.32E−03	5.67E−01	2.55E−01	7.13E−01	1.45E−01	2.37E−01	6.06E−01	5.89E−61	2.90E−01	2.90E−01	
Lano	Estimate	1.08	0.256	−0.6	0.17	0.896	−0.35	−0.221	1.239	0.052	−0.353	
SE	0.273	1.227	0.824	0.991	0.589	0.599	0.108	0.14	0.551	0.54	
p	7.57E−05	8.35E−01	4.67E−01	8.64E−01	1.28E−01	5.59E−01	4.09E−02	6.63E−19	9.25E−01	5.13E−01	
Myev	Estimate	0.449	0.693	0.252	−0.893	−0.666	−0.451	−0.036	1.015	0.216	−0.626	
SE	0.2	0.902	0.601	0.74	0.451	0.44	0.088	0.115	0.396	0.398	
p	2.46E−02	4.42E−01	6.76E−01	2.28E−01	1.40E−01	3.05E−01	6.81E−01	1.10E−18	5.85E−01	1.15E−01	
Mylu	Estimate	2.367	0.824	0.793	0.467	−0.019	−0.563	0.325	0.893	1.04	−1.09	
SE	0.266	1.192	0.812	0.956	0.583	0.597	0.088	0.127	0.541	0.538	
p	4.89E−19	4.89E−01	3.28E−01	6.25E−01	9.74E−01	3.45E−01	2.34E−04	2.30E−12	5.47E−02	4.27E−02	
Myvo	Estimate	−0.508	0.098	−0.328	0.103	1.468	−0.205	−0.075	0.007	1.464	−0.567	
SE	0.225	1.027	0.647	0.864	0.464	0.485	0.109	0.134	0.438	0.44	
p	2.38E−02	9.24E−01	6.12E−01	9.05E−01	1.57E−03	6.73E−01	4.92E−01	9.57E−01	8.32E−04	1.98E−01	
Myyu	Estimate	−3.014	−0.368	0.801	0.309	0.095	−0.431	0.218	0.645	0.015	−1.114	
SE	0.377	1.352	0.809	1.21	0.696	0.71	0.284	0.322	0.635	0.736	
p	1.27E−15	7.85E−01	3.22E−01	7.98E−01	8.91E−01	5.44E−01	4.42E−01	4.51E−02	9.81E−01	1.30E−01	
Note:

For each species-level model (seven models total), we have listed the model estimate, standard error (SE) and p-value for all predictors included in the fixed-effects model structure. Species are identified in the first column using their respective four-letter code (Epfu = E. fuscus, Laci = L. cinereus, Lano = L. noctivagans, Myev = M. evotis, Mylu = M. lucifugus, Myvo = M. volans, and Myyu = M. yumanensis). All models were constructed using the glmmTMB package in R, and interpreted using functions from EcoCountHelper. Models for all species utilized a negative-binomial quadratic error-distribution. Values in the column labelled “Intercept (Site)” correspond with the random-intercept term that incorporates a site-dependence of observations in the models.

Model results are shown in Table 2. Results presented in this format can be difficult to interpret. The “EffectsPlotter” function in the ‘EcoCountHelper’ package allows users to quickly visualize model results as shown in Fig. 3. As mentioned in the Methods section, it is difficult to interpret meaningful effects from scaled predictor coefficients. We used multiple approaches to assess more ecologically-meaningful model results. Utilizing the ‘EcoCountHelper’ “RealEffect” functions, we examined the change in model-predicted change in nightly recorded bat call sequences (as a percent) by specified changes in predictors. This can be done using the “RealEffectText” function which prints a sentence describing the predicted change in nightly bat call sequences for a given change in a single predictor. Using the “RealEffectText” function to interpret the effect of a 10-day increase in ordinal day on M. lucifugus activity results in the sentence: the response variable increases 22.81% (+/-5.91%) for every 10 day increase in the predictor. In this output, the parenthetical value indicates the 95% confidence interval, but the “RealEffectText” function allows users to specify any confidence level for errors surrounding the predicted change in the response variable. Because our analysis included a total of seven parameters of interest for each of the seven species-level models, we chose to use the “RealEffectTab” functions instead of the “RealEffectText” function. The “RealEffectTab” functions accept vectors of models, predictors, and specified changes in each predictor, then generate a table similar to Table 3.

Figure 3 EffectsPlotter function output.

Following the model construction and selection process, we used EcoCountHelper’s “EffectsPlotter” function to visualize scaled parameter estimates with 85% and 95% confidence intervals. Each plot represents a species-level model, with the associated species indicated by the four-letter code above each plot (see Table 2 for species code definitions).

Table 3 Model-predicted responses to changes in predictors of interest.

Predictor classification	Predictor	Unit increase	Epfu	Laci	Lano	Myev	Mylu	Myvo	Myyu	
Anthropogenic	Structure index	0.05	+82.29% (40.79%/123.79%)	+24.46%
(−9.73%/58.64%)	+34.97%
(−12.21%/82.15%)	−19.97%
(−54.43%/14.48%)	−0.63%
(−47.22%/45.95%)	+63.43% (27.83%/99.03%)	+3.24%
(−54.63%/61.12%)	
Proportion cool lights	0.2	−28.30%
(−86.68%/30.08%)	−19.74%
(−65.73%/26.24%)	−17.24%
(−83.68%/49.20%)	+8.25%
(−36.73%/53.23%)	+28.41%
(−36.70%/93.53%)	−9.83%
(−59.02%/39.36%)	+28.74%
(−36.16%/93.64%)	
Brightness index	5	+50.02%
(−158.81%/258.85%)	+29.22%
(−111.16%/169.60%)	+13.04%
(−202.89%/228.98%)	+39.28%
(−93.62%/172.18%)	+48.34%
(−157.43%/254.11%)	+4.82%
(−156.97%/166.61%)	−16.15%
(−271.2%/238.89%)	
Proportion developed	0.2	−29.44%
(−176.37%/117.49%)	−12.11%
(−111.07%/86.84%)	+8.04%
(−133.60%/149.67%)	−33.34%
(−126.69%/60.01%)	+23.65%
(−110.64%/157.95%)	+4.78%
(−111.13%/120.70%)	+15.09%
(−178.78%/208.96%)	
Natural	Elevation (m)	50	+39.45%
(−482.12%/561.02%)	+146.15%
(−198.61%/490.90%)	−45.01%
(−690.08%/600.07%)	−53.73%
(−390.18%/282.72%)	−61.84%
(−701.87%/578.19%)	−29.53%
(−437.02%/377.95%)	−52.19%
(−1033.85%/929.48%)	
Water distance (m)	250	−12.54%
(−38.62%/13.53%)	−9.54%
(−29.96%/10.88%)	−7.95%
(−36.15%/20.25%)	−13.66%
(−33.73%/6.41%)	−22.56%
(−50.62%/5.50%)	−12.45%
(−34.91%/10.00%)	−22.99%
(−63.25%/17.27%)	
Proportion forested	0.2	+19.08%
(−34.58%/72.74%)	−17.35%
(−59.64%/24.94%)	+2.29%
(−57.63%/62.21%)	+9.88%
(−30.35%/50.10%)	+57.22%
(−1.45%/115.89%)	+89.11% (43.80%/134.41%)	+0.65%
(−71.18%/72.49%)	
Note:

Using EcoCountHelper’s RealEffectTab functions, we predicted responses of each modelled species to a specified increase in unscaled (i.e., in the original units) predictors (specified in the “Unit Increase” column), then formatted the table in R. The seven right-most columns each contain the change in species-level activity (represented as a percentage value) as well as the 95% confidence interval range listed parenthetically as “x/y”, where x is the lower boundary of the confidence interval, and y is the upper boundary of the confidence interval. Species are identified by their respective four-letter code (see Table 2 for species code definitions).

We also produced plots that fall outside of the scope of our package due to the necessary specificity of their construction relative to model structure for visually-appealing plots (see associated GitHub repository for the code used to produce these plots). These plots are essentially visualizations of the sentences produced by the “RealEffectText” function, but across the continuum of observed values of each predictor and with predicted nightly call sequences as an absolute metric of bat activity rather than percent change (Fig. 4). Additionally, all other predictors are held constant at their median values in Fig. 4. Both the table generated with the “RealEffectTab” functions and the plots in Fig. 4 were used to gauge the ecological significance of the relationship between predictors and bat activity.

Figure 4 Continuous model predictions for each parameter by individual species.

Plot titles indicate the four-letter code of the bat species with which each plot is associated (see Table 2 for species code definitions). The most appropriate model for each species was identified and validated using functions from EcoCountHelper. All parameters of each model were held at their median observed value except the parameter to be plotted, which was equally distributed throughout its range of observed values. Dark and light shaded areas indicate 85% and 95% confidence intervals, respectively.

Discussion

While the scientific community’s understanding of bat ecology has grown substantially in recent years (Weller, Cryan & O’Shea, 2009), our knowledge is often over-generalized both spatially and demographically which may not provide adequate information for managers to effectively assess the potential consequences of a management decision. The ‘EcoCountHelper’ package presented here is structured with the intent of making model interpretation and visualization more accessible for wildlife managers and ecologists working with ecological count data. We hope that it helps remove a barrier to understanding the effects of locally-relevant landscape features on bats and other wildlife. Using the ‘EcoCountHelper’ package and an associated analytical workflow, we examined the effects of multiple landscape characteristics on bat activity throughout Grand Teton National Park.

In our case study example, we examined the impact of several landscape features on bats in Grand Teton National Park, and found that the presence of porous buildings suitable for roosting had a positive effect on both E. fuscus and M. volans activity. While bat activity and abundance should not be treated as synonymous metrics, a large body of literature exists documenting a multitude of bat species occupying buildings in high numbers (Geluso, Benedict & Kock, 2004; Voigt et al., 2016; Johnson et al., 2019) which is consistent with these findings. Increased bat activity near buildings may be a result of increased bat roosting and foraging near buildings due to decreased predation risk, energetically efficient shelter, or conspecific attraction (Voigt et al., 2016). We also found that an increase in the distance to water had a negative effect on M. lucifugus activity which is consistent with other findings regarding myotid distribution and roost locations (Evelyn, Stiles & Young, 2004; Womack, Amelon & Thompson, 2013). The propensity of M. lucifugus to roost near water may be due to increased insect densities above water (Barclay, 1991) and the use of waterways as travel corridors. Additionally, we found that the proportion of forest in an area had a significant positive impact on M. volans which is consistent with the frequent use of largely forested areas by this species (Baker & Lacki, 2006; Johnson, Lacki & Baker, 2007; Lacki, Johnson & Baker, 2013). While research from Europe suggests that both the presence of artificial light as well as the color of that light may influence bat space use (Stone, Jones & Harris, 2012; Spoelstra et al., 2017), our results here do not suggest the same trends. It is possible that these North American bats are not substantially affected by artificial lights, yet we caution that we had a low number of artificially lit sites (n = 6).

The analytical framework and use of the ‘EcoCountHelper’ R package outlined in this article is meant to serve as a guide for land and wildlife managers to conduct similar research with locally-relevant parameters in mind. There are many analytical solutions for researchers to assess the effect that habitat features and environmental variables have on bats and other wildlife such as spatial autoregressive models (Li & Wilkins, 2014; Ver Hoef et al., 2018), ANOVA (Kalcounis et al., 1999), and models that can account for detectability at the species (i.e., occupancy modelling) (Mendes et al., 2017), site/population (i.e., N-mixture), or individual levels (i.e., mark-recapture) (Kéry & Royle, 2015). We think, however, that the GLMM-based workflow outlined in this article paired with the ‘EcoCountHelper’ package and vignette provides a balance of accessibility, statistical robustness, and research flexibility for analyzing ecological count data. Similarly, while the methods, results and discussion associated with our case study are relatively simple and could be more specifically tailored to provide information about bat habitat selection, the information we presented here can easily be reproduced and modified to suit the needs of other managers while maintaining a relatively simple and robust analysis.

While it was initially written with bat conservation in mind, this package and framework can also be used in tandem for any type of count data (e.g., avian point counts, insect samples, camera trap data, invasive species removal data). We hope that this package provides a clear, succinct, and adaptable analytical framework for managers and ecologists to analyze count data, and facilitates the use of statistically-robust and reproducible methods to inform management actions.

Supplemental Information

Supplemental Information 1 Suplementary Information.

Supplementary information for “EcoCountHelper: an R package and analytical pipeline for the analysis of ecological count data using GLMMs, and a case study of bats in Grant Teton National Park”.

Click here for additional data file.

Supplemental Information 2 EcoCountHelper Vignette.

Click here for additional data file.

We would like to thank Conrad Lucas for assisting in data collection, Shan Burson and Dave Gustine for logistics assistance, and the UW-NPS Research Station for funding, housing and equipment use, and Grand Teton National Park for funding and vehicle use.

Additional Information and Declarations

Competing Interests

Author Contributions

Field Study Permissions

Data Availability

The authors declare that they have no competing interests.

Hunter J. Cole conceived and designed the experiments, performed the experiments, analyzed the data, prepared figures and/or tables, authored or reviewed drafts of the article, and approved the final draft.

Dylan G. E. Gomes analyzed the data, authored or reviewed drafts of the article, and approved the final draft.

Jesse R. Barber conceived and designed the experiments, authored or reviewed drafts of the article, and approved the final draft.

The following information was supplied relating to field study approvals (i.e., approving body and any reference numbers):

The work was conducted under the auspice of Grand Teton National Park and funded under a cooperative ecosystem study unit agreement.

The following information was supplied regarding data availability:

The data and code for data preparation, analysis, figures and tables are available at GitHub: https://github.com/huntercole25/Code-and-Data-for-Bats-in-National-Parks; huntercole25. (2022). huntercole25/Code-and-Data-for-Bats-in-National-Parks: EcoCountHelper Paper Data and Code (EcoCountHelper). Zenodo. https://doi.org/10.5281/zenodo.7246745.

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
