# Peer review of "EcoCountHelper: an R package and analytical pipeline for the analysis of ecological count data using GLMMs, and a case study of bats in Grand Teton National Park"

_PeerJ, doi:10.7717/peerj.14509_

## Round 0.1 · original submission · Major Revisions

Both reviewers agree that your article can be published after major revision. Give special attention to the point made by reviewer 1, who asked for a substantial change in the paper's title and message. I look forward to seeing an improved version.

Reviewer 1 ·

Basic reporting

Authors have to re-organize their narrative.
First of all, the current title is disconnected to the real purpose of this article. In your paper, you actually describe an R package (EcoCountHelper) and an associated analytical pipeline aimed at making GLMM-based analysis of bat passive acoustic monitoring data more accessible. This is goal #1 of your paper. And as goal #2, you field-tested EcoCountHelper and the pipeline in real conditions, in Grand Teton National Park. This is what was presented in this article and this is how the narrative should be. The results on habitat use were superficially presented and, maybe, should be moved to another specific paper where they should receive an indeep analysis.
The current title is inappropriate and should be, for example, "EcoCountHelper: an R package and analytical pipeline for the analysis of bat acoustic data and landscape variables". A title like that expresses more correctly what you have presented.
Taking that into consideration, I missed more graphical material (figures and a workflow chart, for example) for the presentation of EcoCountHelper. You claim that most potential users for this kind of package lacks programming or even statistical skills, therefore, you will have to be very didactic in the explanations. Figures, schemes and workflows may be very usefull for making your package more user-friendly and attractive. I recommend you investing more on a better presentation of the necessary steps for running EcoCountHelper, for example. This is very necessary considering that you have a long methodology section basically describing the functioning of the package.
If you decide to maintain the analysis on habitat use, then you will have to improve that part a lot, because that specific part of the paper was poorly and incompletely presented at all. Similarly, the discussion of the habitat use in its current form is far from acceptable. In its current form, the paper should be rejected. However, I am suggesting you to focus on the description and testing of the package so you could put less "weight" on the habitat use analysis, indicating that it would be published elsewhere.

Experimental design

Correct for testing the R package presented. But the superficiality of the discussion on habitat use compromises the overall quality ofthe manuscript.

Validity of the findings

Problematic. The habitat use component of the paper was poorly and incompletely presented, as well as the discussion of that specific part of the results. The R package is useful and should be the main focus of this paper. A more specific and in-depth approach to the use of habitat by bats should be the subject of another article.

Reviewer 2 ·

Basic reporting

The manuscript entitled "Bats in National Parks: An analysis framework and case study for understanding bat habitat use in protected areas" is interesting and the authors make an overview of other packages to compose a new one that can facilitate statistical analysis in the area of biodiversity. The authors argue that this package can be used by park managers in their analysis, mainly to try to understand habitat use and anthropogenic effects on fauna. The text is clear and well written. However, it needs some clarification on the methods, as well as verifying that some added variables (moonlight) cannot lead us to a wrong interpretation, since there is a bias in the data. In the attached PDF are some other criticisms and suggestions that can improve the text.

best,
Reviewer.

Experimental design

Ok

Validity of the findings

Ok

Annotated reviews are not available for download in order to protect the identity of reviewers who chose to remain anonymous.

---

## Round 0.2 · accepted · Accept

Please work with our production team to have your paper published. Congratulations again!

Reviewer 2 ·

Basic reporting

I read again the manuscript entitled "EcoCountHelper: an R package and analytical pipeline for the analysis of ecological count data using GLMMs, and a case study of bats in Grant Teton National Park". The authors have made all the modifications and answered all the questions. Now, I suggest publishing the manuscript. Congratulations on the study and effort in correcting the text!

Experimental design

No comments.

Validity of the findings

No comments.

Additional comments

No comments.